# Thermoplastic Composite Materials Approach for More Circular Components: From Monomer to In Situ Polymerization, a Review

**Marco Valente** [1,2,*] **, Ilaria Rossitti** [1] **, Ilario Biblioteca** [1] **and Matteo Sambucci** [1,2]

[1] Department of Chemical Engineering, Materials, Environment, Sapienza University of Rome, 00184 Rome, Italy; ilaria.rossitti@uniroma1.it (I.R.); ilario.biblioteca@uniroma1.it (I.B.); matteo.sambucci@uniroma1.it (M.S.)

[2] INSTM Reference Laboratory for Engineering of Surface Treatments, Udr Rome Sapienza, Sapienza University of Rome, 00184 Rome, Italy

\* Correspondence: marco.valente@uniroma1.it

**Abstract:** To move toward eco-sustainable and circular composites, one of the most effective solutions is to create thermoplastic composites. The strong commitment of world organizations in the field of safeguarding the planet has directed the research of these materials toward production processes with a lower environmental impact and a strong propensity to recycle the polymeric part. Under its chemical properties, Nylon 6 is the polymer that best satisfies this specific trade-off. The most common production processes that use a thermosetting matrix are described. Subsequently, the work aimed at investigating the use of thermoplastics in the same processes to obtain comparable performances with the materials that are currently used. Particular attention was given to the in situ anionic polymerization process of Nylon 6, starting from the ε-caprolactam monomer. The dependencies of the process parameters, such as temperature, time, pressure, humidity, and concentration of initiators and activators, were therefore investigated with reference to the vacuum infusion technique, currently optimized only to produce thermosetting matrix composites, but promising for the realization of thermoplastic matrix composite; this is the reason why we chose to focus our attention on the vacuum infusion. Finally, three production processes of the polymeric matrix and glass fiber composites were compared in terms of carbon footprint and cumulative energy demand (CED) through life-cycle assessment (LCA).

**Keywords:** thermoplastic resin; casting; chemical properties; anionic polyamide 6; vacuum infusion; composite LCA

## 1. Introduction

The use of composite materials is rapidly expanding in many industrial sectors thanks to their ability to combine different properties and, thus, become extremely versatile. In recent decades, the development of eco-sustainable composites has involved the study of glass/polymer [1], carbon/polymer [2], metal/polymer [3], polymer/polymer [4], ceramic/polymer [5], and concrete/polymer [6] interactions for the optimization of the final mechanical properties and the inclusion of these products in the perspective of environmental and economic circularity.

The will to produce a thermoplastic matrix composite lies in its intrinsic nature, which is opposite to that of the thermosetting matrix, that is, the possibility of recycling the material.

Due to the higher toughness of the matrix, TPCs potentially offer a higher resistance to fatigue than their thermoset-based counterparts that are currently used, such as fiber-reinforced epoxies and vinyl esters. The current most widely applied manufacturing process, vacuum infusion, is the preferred choice to produce large composites. This process

utilizes a low-viscosity resin that is injected into a mold with pre-placed dry fibers, followed by a curing step. Since the viscosity of thermoplastic polymer melts is very high, reactive processing is required: a low-viscosity monomer melt is injected between the fibers, together with an activating system, if required, followed by in situ polymerization of the monomer to form a linear polymer with a sufficiently high molecular weight [7].

With this type of processing, it is possible to manufacture larger and thicker parts with a higher level of integration and complexity than is possible with the traditional melt processing of thermoplastic composites. The resin used is anionic polyamide-6 (APA-6) with a catalyst system specifically formulated for composites. At temperatures between 150 and 190 °C, the monomer melt has a water-like viscosity and polymerizes into highly crystalline polyamide-6 in 45 min [8]. APA-6 composites can therefore be processed similarly to thermoset composites through, for instance, vacuum infusion or resin transfer molding (RTM), while maintaining weldability and recyclability. The polymer morphology that is obtained during reactive processing of APA-6 results in a much higher modulus and strength than melt-processed PA-6 and reduces the property degradation due to moisture absorption. The material costs of the monomer and catalyst system are lower than those of polyamide-6 and commonly used epoxy resins; the final processing costs of the developed technology are currently under investigation [8]. The fallback choice of Nylon 6 as a thermoplastic matrix is due to the characteristics of the polymer. Its macromolecules can reorganize in space to obtain a crystalline structure enough to obtain a mechanical strength better than that of other thermoplastics, such as, for example, high-density polyethylene (HDPE) and polypropylene (PP).

These composites have achieved good popularity in fields such as automotive, aerospace, and wind power, presenting themselves as low-weight and sufficiently resistant products to withstand several types of loads. The basic concept is that the use of a vacuum infusion process as a technique to produce composites has always involved the use of thermosetting polymers as a matrix; however, the purpose is to attempt the implementation of this technique toward the use of thermoplastic matrices. The benefits are as follows:

I.    Using a vacuum as a driving force for impregnating the fibers does not require the application of high-pressure values, offering the possibility of obtaining dimensions and thicknesses similar to those of thermoset matrix composites.

II.   The in situ polymerization of the thermoplastic matrix around the fibers allows us to establish and form a strong chemical bond at the interface, which is much more difficult to obtain with traditional spindle processing methods; this allows for an increase in fatigue performance of thermoplastic composites.

III.  The vacuum infusion process is commonly applied to the production of wind turbine blades and, consequently, does not require the introduction of entirely new processes and technologies.

IV.   The possibility of significantly reducing costs [9].

However, attention must be paid to some key process parameters, including the polymerization temperature, since, in general, an increase in it results in a higher polymerization rate and a higher degree of branching; on the other hand, it also results in a lower degree of crystallinity [10].

## 2. Anionic Polymerization of PA-6 from the Epsilon-Caprolactam Monomer

The anionic polymerization process was studied and optimized for a long time before being included in the development of composite materials in which, starting from the 2010s, it began to interest the scientific and industrial community, not only for the excellent mechanical and thermal properties of the polymer produced [11,12], but also for their thermal, mechanical, and chemical recyclability [13]. Among the polymers that show these characteristics, the one of greatest interest in the field of composites is PA-6, thanks to its low viscosity, which makes it suitable for infusion into molds [14]. PA-6 belongs to the family of polyamides which, based on the length and chemical organization of the chains, also includes polymers such as PA-6-6, PA-11, PA-12, and PA-4-6 [15].

PA-6 can be produced mainly hydrolytically, cationically, and anionically; in this third case, the process, which is faster, more stable, and easier to control, is advantageous for engineering [16–18]. The anionic polymerization of PA-6 occurs at relatively low temperatures (140–180 °C), starting from the cyclic monomer of epsilon-caprolactam, which, in the presence of a certain concentration of initiators and activators, develops chains linked together by a semicrystalline structure which confers the desired properties.

The anionic polymerization of the ring-opening of caprolactam (as in other lactams) follows an activated monomer mechanism; that is, the chain growth reaction proceeds from the interaction of an activated monomer (lactam anion) with the end of the chain in growth. Anion attack is the dominant step for speed in propagation. The other feature of this mechanism is that the activated monomer is regenerated after each unit growth reaction. In the literature, there are many compounds capable of catalyzing this reaction, such as alkali metal hydroxides, alcoholates, carbonates, Grignard reagents, alkylaluminiums, alkalialuminium hydrides and their partial or total alkoxides or their lactam salts, quaternary ammonium salts of lactams or of other compounds, and guanidinium salts of lactams. The most used activators are *N*-acyllactam, monofunctional *N*-acetylcaprolactam and difunctional hexamethylene-1,6-dicarbamoylcaprolactam [16,19]. The most common activators in the articles reviewed in this work are sodium caprolactamate and caprolactam magnesium bromide, both preserved in a flake form.

The reaction mechanism, schematized in Figure 1, has been studied and modeled respecting the mass and energy balances, as well as the kinetics of polymerization and crystallization [20]. From this model, useful relationships were found for calculating the induction time and the initial crystallization temperature. Based on this, it was possible to optimize and predict the reaction kinetics in specific production processes. It has been confirmed that the parameters that most influence the anionic polymerization are the initial crystallization temperature and the concentration of activators and initiators [7,10]. The study and optimization of the effects of these parameters on the degree of polymerization and crystallization reflect the production of PA-6 with adequate thermal, mechanical, physical, and chemical properties.

**Figure 1.** *Cont.*

**Figure 1.** Initiation (**a**,**b**) and propagation (**c**) of the AROP of lactams, schematically. Note: x = 5 and 11 for CL and LL, respectively. Reprinted from Reference [18], under open-access license.

In addition to the performance aspect, industries require materials that come from green processes. Under certain conditions, the anionic ring-opening polymerization reaction proceeds at high conversion rates and with negligible wastes of matter. Beyond being green, this process has epsilon caprolactam as its reagent, which can be synthesized from renewable sources [21–23] and produces a polymer (PA-6) that can be returned to the starting monomer and is, therefore, recyclable.

## 3. In Situ Polymerization

The possibility to have a powdered reagent that can synthesize PA-6 in minutes has enabled the massive development of many techniques, including in situ polymerization [24]. In the production of nylon objects starting from polymer, there are criticalities in the rheology of the fluid due to its high viscosity [25]. Starting from the monomer in the form of flakes, however, it is possible, by heating, to obtain a much less viscous monomeric fluid that polymerizes in a semi-crystalline form in a range of 140/180 °C [12]. In this case, the temperature influences the degree of polymerization and crystallization at the same time. At high temperatures, the polymer produced has a high molecular weight but a high amorphous fraction and vice versa [12]. For most applications and experiments, the optimal process temperature is 140/150 °C [26]. For each technique, the optimization of the process was investigated with the right temperature, the right concentration of activators and initiators, and ensuring the absence of water in the entire process.

With the use of powerful statistical tools, different kinetic models of crystallization have been identified [27,28]. Specifically, for the anionic polymerization of PA-6, the isothermal crystallization model is more recognized. The predictability of the polymerization/crystallization process makes it possible to optimize production processes. The fluid dynamic characteristics of the polymerizing caprolactam during mold filling were also investigated [25]. Thanks to these models, it was possible to build time–temperature diagrams for the ring-opening anionic polymerization process of great applicational utility, on a par with the time–temperature–transformation (TTT) and conversion–temperature–transformation (CTT) digraphs for thermosets [29].

### 3.1. Casting

Casting is one of the oldest in situ polymerization techniques of thermoplastics and thermosets [30,31]. With this technique, it is possible to produce large objects in an economical and energy-efficient way. This technique, still used today, has undergone improvements thanks to the addition of additives that improve the product. PA-6 products made with the casting technique, in fact, show low impact resistance and low elongation at break values. To improve these properties, some researchers have investigated the use of additives such as UHMWPE powder and HTP [32], which provide an increase in the plastic component, while reducing the degree of crystallinity. An advantageous alternative is the use of copolymers [33], with which it is possible, by balancing the concentrations of different monomers,

to obtain products suitable for applications that require high impact resistance. However, the Nylon-6 copolymers appear to have slightly lower tensile strength values. Furthermore, thanks to the use of software for simulating the process, it was possible to obtain conversion degrees of 95%, a molar mass of 40–50 kDa, and a crystallization percentage of 40–44% [34].

### *3.2. Centrifugal and Rotational Reactive Molding*

The centrifugal and rotational reactive molding techniques, already previously used starting from polymer, involve the simultaneous polymerization and forming of PA-6 starting from the monomer [35–38]. Centrifugal molding is mainly used to produce components with axial symmetry geometry, while the rotational molding technique is used for more complex geometries in which the product is made in a single piece and without welding. During their rotation, the molds go through ovens which favor the polymerization reaction with subsequent cooling of the polymer. The simulation of this process (isothermal crystallization model) made it possible to define the optimal parameters of crystallization start time and temperature, cycle times, and rotation speed. The most critical aspect of these techniques is the control of viscosity during the process to obtain homogeneous thickness and performance throughout the product. This parameter can be predicted through the graphs of isoviscosity as a function of time and temperature during polymerization. These graphs are determined at the constant composition of monomer/catalyst/activator and indicate the processing regions in which the associated operating parameters guarantee optimal viscosity values in the range between 1 and 0.05 Pa·s [36].

Compared to the classic techniques starting from molten polymer, the use of the reactive monomer leads to the formation of products with better tensile properties: the Young's modulus improves from 750 to 1560 MPa, yield stress goes from 62 to 80 MPa, and elongation to break goes from 32% to 64% [36], see Table 1.

**Table 1.** Material characteristics and processing parameters when comparing classical versus reactive processing. Adapted from Reference [36].

| Rotational Molding Technique of PA6 | Classical | Reactive |
|---|---|---|
| Temperature | T~240 °C | T = 150 °C |
| Cycle time | t > 40 min | t = 15–20 min |
| Degree of crystallinity (%) | 28 | 49 |
| Intrinsic viscosity (dL/g) | 1.07 | 7 |
| Molecular weight (g/mol) | 30,778 | 182,594 |
| **Tensile properties** | | |
| Young's modulus (MPa) | 750 | 1560 |
| Yield stress (MPa) | 62 | 80 |
| Elongation at break (%) | 32 | 64 |

### *3.3. Reactive Extrusion*

The reactive extrusion process, investigated since the 1980s by Michaeli [39], allows the continuous production of PA-6, starting from the caprolactam monomer, which is preheated and subsequently poured into a twin-screw, in which it polymerizes and is extruded. Moreover, for the extrusion, it was possible to optimize the operating parameters thanks to numerical simulations of the reactive process [40]. The model that defines this process relates the evolution of the monomer conversion to the reaction time, screw speed, monomer feeding speed, and the different screw configurations [41].

### 4. In Situ Polymerization for Composites

Due to the extremely low viscosity of the cyclic lactams and the superior mechanical properties of the polymers obtained from them, these materials have great potential for application in different liquid composite molding (LCM) techniques. Unsurprisingly, extensive academic research has been conducted over the past 20 years to investigate

possible industrial applications of anionically polymerized thermoplastic composites (TPCs) reinforced with glass, carbon, aramid, or natural fibers [18]. It is important to remember that enormous progress has been made in the development of machinery and materials [42]. There are different types of reactive processes for obtaining Nylon-6 composites; some of the main technologies are presented below.

### 4.1. Reactive Injection Pultrusion

The thermoplastic reaction injection pultrusion provides an alternative for the preparation of thermoplastic composites with high fiber content. Pultrusion is one of the most cost-effective and energy-efficient methods of manufacturing continuous fiber-reinforced composite profiles. Over the past 70 years, thermosetting resins have dominated the pultrusion industry, owing to their rapid curing and effective impregnation, along with low viscosity. However, thermosetting resins have several disadvantages. They are brittle, sensitive to impact, and cannot be recycled. In addition, volatile compounds would be released during the pultrusion process of many thermosetting resins, and this is contrary to environmental protection policies. Thermoplastic polymers completely avoid the above shortcomings and offer improved impact strength, damage tolerance, toughness, and reparability [43]. Nevertheless, a relative high viscosity (100–10,000 Pa·s) of molten thermoplastics [44] limits the impregnation ability, resulting in the poor quality of the pultruded products. Several impregnation techniques for thermoplastic pultrusion have been developed by researchers. One way was to put the polymer and fibers in intimate contact prior to the final molding step. Pre-impregnated materials, such as prepreg tapes, commingled fibers, and powder coated towpregs, had been well-developed for pultrusion processes [45–48]. However, there was still a large gap in achieving good impregnation with high fiber content between thermoplastic polymers and thermosetting resins [43].

In order to solve this problem, reactive processing of thermoplastic composites with anionic ring-opening polymerization of polyamide-6 (PA-6) has been developed by researchers [7,10]. The impregnation of high content fibers can be easily achieved through caprolactam (monomer of PA-6) with extremely low viscosity (5 m Pa·s) [10]. Luisier et al. developed a pilot reactive injection pultrusion line in the base of anionic polymerization of polyamide-12 (PA-12) [49]. Epple et al. successfully prepared thermoplastic composites with anionic PA-6, using a pultrusion process, but the optimization of process condition, properties, and microstructure of pultruded composites were not reported in detail [50].

### 4.2. Infusion Techniques

The infusion techniques described in this section are already optimized for thermosets, while, for thermoplastics, researchers are still trying to implement them, as there are some difficulties.

Three reactive processes are introduced: structural reaction injection molding, resin film infusion, and vacuum infusion (see Figure 2). Reactive processes for the manufacturing of short-fiber-reinforced composite parts, such as reinforced reaction injection molding (RRIM) [28,51–53], are discussed in the literature.

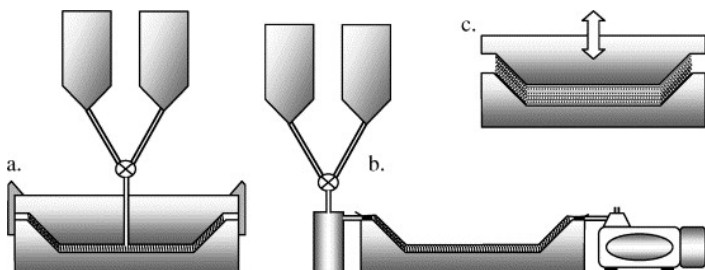

**Figure 2.** Schematic representation of three reactive processes for manufacturing of thermoplastic composites: (**a**) structural reaction injection molding (SRIM), (**b**) vacuum infusion (VI) and (**c**) resin film infusion (RFI). Reprinted from Reference [54], under open-access license.

### 4.2.1. Structural Reaction Injection Molding (SRIM)

The mixture is infused onto the dry fiber bed, which is placed between two solid mold halves, at high pressure (8–10 bar) (see Figure 2a) [24,55–67]. The process is closely related to Resin Transfer Molding (RTM) of thermoset composites [43]. To obtain a proper fiber impregnation, 1 Pa s is generally regarded as the maximum viscosity limit of the reactive mixture [68]. At the end of the polymerization, the composite part can be demolded. As the size of the product increases, the clamping force required to keep the mold closed during resin injection also increases. As a result, the dimensions of parts produced via SRIM are limited. The reactive material systems are separated into two material feeds to prevent premature polymerization. Just before entering the mold, material feeds from two tanks are mixed and polymerization begins. The great advantage of SRIM is the fast cycle time, and the high tooling costs are the main disadvantage.

#### Injection Molded Thermoplastic Composites

Injection molding, which allows for the fast, low-cost production of composite parts, is, however, a commonly used technique to produce short-fiber-reinforced thermoplastics. Their applications are growing rapidly, because of the good feature of the net-shaped components with good dimensional accuracy and relatively short cycle time [69]. Short-fiber reinforcement of thermoplastics materials is extensively used because it results in composites with high impact resistance, good dimensional stability, a high strength-to-weight ratio, and good thermal insulation properties of the molded parts comparable to unreinforced plastics and some conventional metal alloys. However, the mechanical properties of short-fiber composites are not as good as composite with continuous reinforcing fibers [70]. According to Fu et al. [71], the specific work of fracture or the fracture toughness of short-fiber composites is caused by all of the following failure mechanisms: (1) fiber–matrix interfacial debonding, (2) post-debonding friction, (3) matrix plastic deformation, (4) fiber plastic deformation, (5) fiber fracture, (6) matrix fracture, and (7) fiber pull-out, and it is closely related to the fiber volume fraction and the fiber length. In general, the failure mechanisms depend not only on the properties of the constituents but also on the bonding efficiency across the interface, as well as the fiber volume fraction, the fiber length, and the fiber orientation.

This technology currently allows for the production of PA-6 matrix thermoplastic composites reinforced with short glass fibers such as polyamide PA-6 Nylon composite pellets. It is reiterated that, to produce this type of material, the process does not start from the monomer, but directly from the polymerized thermoplastic, which, in turn, is mixed with the short fibers and is molded by injection molding.

To manufacture long-fiber composites and ensure a good impregnation, starting from the monomer is the only way. This also brings into play the possibility to impregnate non-woven fabrics and move toward an increasingly green direction, since, in contrast to the fabric, there is the chance to create composites with recycled fibers (TNT) and a recyclable matrix. This is the main reason why, in this review work, we have focused in detail on the vacuum infusion technology, which we discuss in detail in Section 5.

### 4.2.2. Resin Film Infusion (RFI)

Fabrics already pre-impregnated by the mixture are used with inhibition of reactivity during the handling and storage phases. This is a film of resin superimposed on the fabric, generally referred to as thermosetting matrices, such as epoxy resin. The polymerization will take place through the application of heat and pressure (see Figure 2c) [72].

### 4.2.3. Vacuum Infusion

In a common vacuum infusion process for polymer composite [73] dry fiber fabrics are placed on a solid mold half, which defines the part geometry (see Figure 2b) [7,73–78]. Among the processes just described, vacuum infusion turns out to be one of the most effective techniques that allows treating high volumes with simpler and even cheaper

equipment. Study and analysis of the operating conditions of the process are discussed in more detail in Section 5.

*4.3. Thermoplastic Resin Transfer Molding (T-RTM)*

In resin transfer molding (RTM), the reinforcing material is preplaced in the mold and impregnated with the matrix material by pressure or vacuum. This technology has made it possible to reduce manual work, but, despite this, the cycle time and, for most of the resins, the curing time are still long. Furthermore, the recycling of crosslinked matrix composites on an industrial scale is still in its infancy [79]. T-RTM was developed to overcome these problems. The technology is based on RTM, which was previously used for crosslinked composites, but, in this case, is used for reactive thermoplastic polymers. In reactive processing, polymerization takes place starting from monomers with the addition of an initiator and activator. Monomers have a low molecular mass, which leads them to have a viscosity similar to water, so they can easily impregnate the dry reinforcement material. During reactive processing, a chemical change occurs, but the result is a thermoplastic polymer without crosslinking [80].

High reinforcing fiber content can be achieved by this process, and no post-processing of the finished product is required [81].

## 5. Vacuum Infusion Molding

This is the process on which we have mainly focused because, among the infusion techniques, it is the one that is currently taking hold from a commercial point of view. This is justified by the fact that companies such as Bruggemann have set up products precisely targeted for vacuum infusion technology [82]. The vacuum infusion process is presented as a variant of the common RTM process, allowing for the treatment of more important volumes with cheaper equipment. In the literature, the vacuum infusion process takes place inside a mold consisting of a solid part at the base and a flexible part of the polymeric film as a closure (vacuum bag), which is held hermetically by a sealing tape.

Although the impregnation rate is lower than at SRIM, the fact that atmospheric pressure is sufficient for the clamping of the mold means that the maximum part size obtainable is limited only by the pot-life of the reactive system. After mixing the contents of the two material tanks, the reactive mixture (maximum viscosity: 1 Pa s) is dispensed into a buffer vessel, which is necessary to separate the pressure required for the delivery and the vacuum necessary to facilitate the infusion. The great advantages of vacuum infusion are the virtually unlimited size of the parts that can be produced and the low-cost equipment due to the low pressures involved. The disadvantages are related to the flexible mold half, which often can be used only once and leads to poor surface quality on one side of the product [83].

With this method, it is also possible to produce large components such as turbine blades, which, with some methods, such as RTM, would require high costs to adapt the process to the required dimensions.

This process is applied to thermosetting and thermoplastic matrices; however, most productions mainly involve the first category. In fact, there are differences at the base [54]:

- The rate of polymerization increases with increasing temperature; this true for both thermoplastic and thermoset resins. When processing reactive thermoplastics of a semi-crystalline nature, however, it must be borne in mind that crystallization is adversely affected by temperature [10].
- Some reactive thermoplastic materials, such as PA-6, PA-12, and PBT, have a melt viscosity which is an order of magnitude lower than common thermosetting resins. Consequently, the capillary forces that occur during the impregnation of the fiber preform are significant and constitute a potential source of voids and runner formation [49,56].
- The performance of composites is not only determined by the fibers and the matrix, but also by the fiber–matrix interphase. To improve this bond, glass fibers, for example, are

usually coated with silane coupling agents: bifunctional compounds with the ability to bond both with the fibers and with a polymeric matrix of choice. An incompatible coupling agent results in weak interphase or even prevents the polymerization of reactive resins. Coupling agents have been developed for several thermosetting composite resins and for thermoplastic composites produced by melt processing. Specific coupling agents for reactive processing of thermoplastic composites have not been developed yet but have recently become a topic of interest [1,84].

## 5.1. Examples of Vacuum Infusion Systems

Among the works developed in this field, we find the research of Ben et al. [85], in which a new method was presented to produce fiber-reinforced thermoplastic matrix composites (FRTPs), exploiting the in situ polymerization of caprolactam; NaCl (C10) was used as initiator, and HDCL (C20) was used as an activator. The reinforcements used are carbon fiber twill fabrics and plain glass fiber fabric, respectively, 13 and 15 layers. The thickness of the carbon fabrics is 0.22 mm, and the density of the fabric is equal to 12.5 tows/25 mm in both weft and warp directions; the thickness of the glass fibers, on the other hand, is 0.21 mm, with a density of 20 tows/25 mm, always in both directions. The carbon fibers were washed with acetone to remove the carboxylic acid of the coupling agent, as it negatively affects the hardness of caprolactam. Instead, the glass fabrics were processed with the insertion of silane as sizing.

The fabrics were stacked inside the rigid metal mold and heated to precise temperatures of 120, 140, 160, and 200 °C. At the same time, the internal part of the mold was depressurized thanks to the use of a vacuum pump up to a value of −90% gauge pressure. After this, the solution containing the monomer was mixed with the initiator and activator at 110 °C and inserted inside the mold maintained at a specific temperature. After infusion, the mold and the internal solution were kept at the same temperature for the time necessary for polymerization to take place. Once the polymerization was complete, the PA6 matrix composite was removed from the mold.

As a result, there are no voids, thanks to the low viscosity of the caprolactam, with the same dimensions for both reinforcements equal to 710, 630, and 3 mm for length, width, and thickness. The number of reinforcements is 15 for glass fabrics and 13 for carbon fabrics, with volume fractions of 42% and 49%, respectively [85].

Another description of an experimental vacuum infusion plant was proposed by Yan et al. [86].

In the mixing vessel, there is the entry of nitrogen as an inert gas and the outlet channel for the reactive mixture. The reagents used are caprolactam, C20 as activator, difunctional hexamethylene-1,6-dicarbamoylcaprolactam (2 mol/kg of caprolactam), C10 as initiator, and sodium caprolactam (1 mol/kg concentration in caprolactam). For the drying process, the reagents were dried for about 24 h before the test at 50 °C, while the fiber fabric, and for this test, glass fibers (S-glass, 400 $g/m^2$), were dried for 120 min at 120 °C. For the mold, a polyimide film was chosen as the vacuum bag, and another polyimide film of a different type was chosen as peel ply. Through these main examples, it was possible to design, from the previous work, a pi-lot plant that would allow for us to mirror the main components, such as the melting, mixing, and passage, in the mold through a vacuum machine.

It is interesting to describe a further method, the SCRIMP method (Seemann Corporation Resin Infusion Molding Process, patented by Seemann Corporation), which is an improvement of the standard VARTM (Vacuum Assisted Resin Transfer Molding) [87], as its objective is to reduce the impregnation times of the fiber bed by of the reactive mixture. In this method, a layer of porous plastic, called infusion media, is interposed between the fiber bed and the vacuum bag. This layer has a 2–3 times greater permeability than the preform of fibers, thus favoring a rapid dispersion of the resin in the upper part of the reinforcement: in this way, the reactive mixture penetrates mainly in the direction of the thickness, rather than through the cross-section, running along the entire length of the fabric. In this way, the infusion operation is faster, also requiring a lower pressure difference

as the driving force [88]. It is possible to produce larger components than VARTM with the added advantage of having shorter times, also reducing the risk of obtaining voids caused by the absence of impregnation.

A disadvantage of using vacuum infusion with a vacuum bag is in the moment of removal from the mold: two surfaces of the composite will be visible, one smooth and well-defined on the side in contact with the metal plate, and the other, in contact with the polymeric film, more wrinkled and less defined [83].

### 5.2. Analysis of the Operating Conditions of the Process

5.2.1. Effects of the Polymerization Temperature on the Properties of APA-6

The choice of the polymerization temperature is essential for obtaining specific properties for the polymer; indeed, various characteristics depend on it both from a microscopic and a macroscopic point of view. The choice of this temperature must be carefully considered. Since we are in the presence of a semi-crystalline polymer, the polymerization and crystallization will occur simultaneously. At high temperatures, there is an increase in the reaction rate, but a decrease in the crystallization rate. At high temperatures, there is an increase in the reaction rate, but a decrease in the crystallization rate. If the temperature is too low, for example, below 130 °C, the crystallization rate is so high that the reactive groups intended for polymerization are trapped inside the crystals in the growth phase. From the following photos, it emerges that, at high temperatures, a homogeneous translucent polymer is obtained, while at lower temperatures, a surface with more opaque parts is obtained [83].

5.2.2. Effect of the Polymerization Temperature on the Molar Mass

Figure 3 shows how the average molar mass of the infused polymer rapidly increases as the temperature of the mold increases.

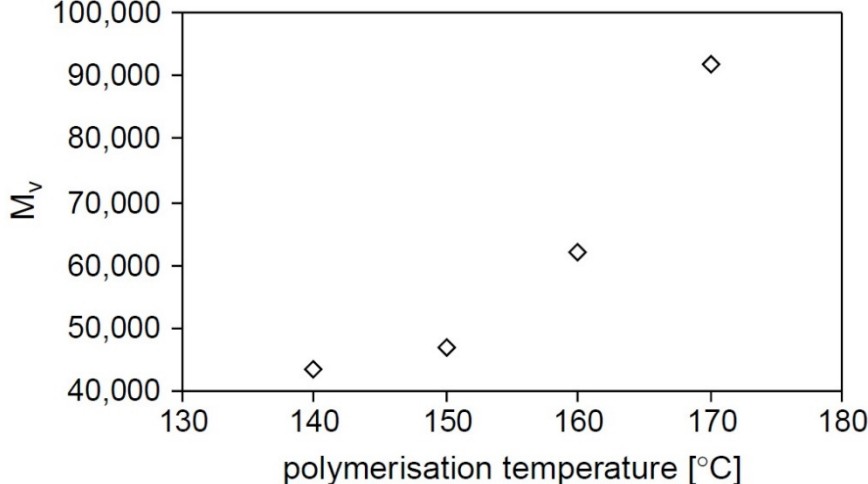

**Figure 3.** Viscosity average molar mass for various polymerization temperatures. Reprinted from Reference [10], under open-access license.

This increase is explained by the fact that the activator was achieved by blocking the diisocyanate by the caprolactam rings. The latter are not stable with temperature, and, if above 160 °C, the activator unlocking reaction occurs. While, in the blocked configuration, it allows the growth of the macromolecule, in the unblocked form, it can react with the active groups of the neighboring chains to implement branching. This is because, once the chain has started to grow, the carbamoyl group becomes an acyllactam group and is, therefore, unable to form an isocyanate: this implies the impossibility of branching [89]. What favors a greater branching are the high concentrations of activator, since there are more active sites

from which to start the growth of a chain, and the use of a slower combination between activator and initiator, which allows the branching to have more time to happen [10].

If there is moisture, the activator function can also be inhibited. The reactive isocyanate group (I) can react with a water molecule to form an unstable carbamic acid (II), which decomposes into a primary amine and carbon dioxide (III). The amine can further react with another isocyanate group and form urea (IV), thus blocking the growth and branching process. Problems relating to the presence of humidity also occur for the initiator, leading to the formation of more complicated compounds that will not take part in the polymerization reaction [90].

### 5.2.3. Effect of the Polymerization Temperature on the Degree of Conversion

The final conversion degree decreases as the polymerization temperature increases, see Figure 4. There are two reasons that explain this behavior: the first reason is due to the inversion of the reaction equilibrium toward the monomer, and the second concerns the decrease in the degree of crystallinity allowing the diffusion of the monomer toward the outside.

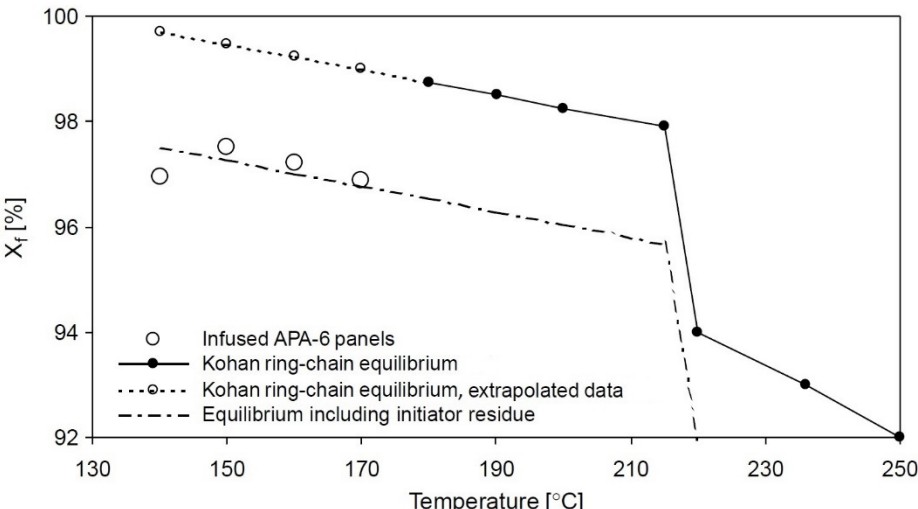

**Figure 4.** Final degree of conversion (t = 30 min) of the infused polymer panels for various polymerization temperatures and Kohan ring–chain equilibrium data. Reprinted from Reference [10], under open-access license.

At 140 °C, however, the crystallization rate is so high that it is able to block the monomer within the growing crystallite. This trapped caprolactam not only reduces the measured conversion, but also causes the whitish coloring of the crystals thus formed, as seen above. At high temperatures, on the other hand, the conversion drops further as the branching process is favored, and this is to the detriment of the growth process of the macromolecule [10].

### 5.2.4. Effect of the Polymerization Temperature on Crystallinity

As the polymerization temperature increases from 140 to 170 °C, the degree of crystallization gradually decreases from 42.2 to 33.2% [10] (see Figure 5).

This is evidenced by the fact that the melting temperature tends to decrease and, consequently, a smaller amount of heat flow (mJ/g) will be required to bring melting the crystalline region of the polymer.

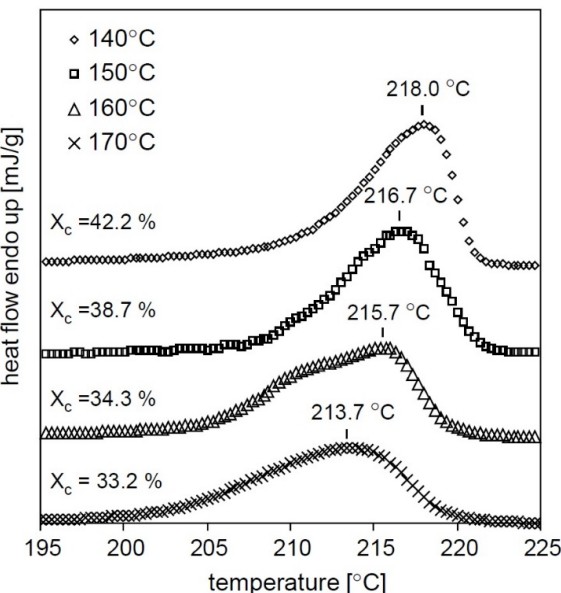

**Figure 5.** Degree of crystallinity and polymer melting point for various polymerization temperatures as measured by DSC (curves are shifted up for clarity). Reprinted from Reference [10], under open-access license.

A first explanation is connected to the fact that, the more the temperature increases, the more the mobility of the macromolecules is favored, resulting in a lower possibility of forming crystallites [83]. A second explanation is that branch points disturb the formation of crystals and, therefore, reduce the degree of crystallinity [91].

As a confirmation of what has just been said about the branching process, Figure 6 compares the degree of crystallinity at various polymerization temperatures for two different formulations of activator and initiator (1.2 mol% HDCL versus 0.6 mol% HDCL). Below the de-blocking temperature (160 °C), no branching occurs, and the higher concentration of activator leads to a greater number of polymer chains and, consequently, a lower molecular weight. As already explained, a lower molecular weight leads to a higher degree of crystallinity. Above the release temperature, however, a higher concentration of activator causes a significant increase in branching, which causes a lower crystallinity.

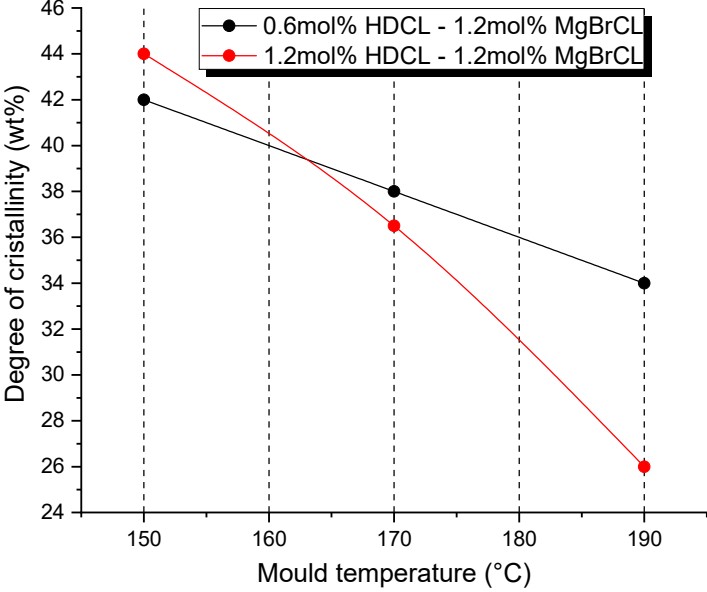

**Figure 6.** Degree of crystallinity at various mold temperatures for two different activator concentrations. Authors' own figure based on the data from Reference [83].

### 5.2.5. Effect of the Polymerization Temperature on the Polymer Melting Point

An increase in the polymerization temperature from 140 to 170 °C corresponds to a decrease in the melting temperature of about 4 °C. The presence of residual caprolactam and the defects present in the crystals induced by the ramifications cause a decrease in the melting temperature, as lower energy (mJ/g) is required to break a crystallite. Ricco et al. reported melting point depressions between 3.5 and 5 °C per wt.% residual caprolactam. Another possible theory links the formation of branches with a transition of the structure of the crystals, since, during the polymerization phase, two different types of crystals appear: the one with $\alpha$-structure (Tm = 220 °C) and the one with $\gamma$-structure (Tm = 214 °C). For anionic PA-6, it is reported that, due to excessive branching, the $\gamma$ content can increase, hence reducing the melting point [10].

### 5.2.6. Effect of the Polymerization Temperature on the Tensile Properties

As expected, according to Table 2, the polymerization temperature also influences the final mechanical characteristics of the product, see Figure 7. Between 150 and 170 °C, there is a decrease in both the elastic modulus and the tensile strength because it goes toward a lower formation of crystals.

**Table 2.** Tensile properties of dry as molded (DAM) APA-6. Reprinted from Reference [10], under open-access license.

| Polymerization Temperature (8C) | Young's Modulus (GPa) | Yield Stress (MPa) | Yield Strain (%) | Maximum Stress (MPa) | Maximum Strain (%) | Strain at Break (%) |
|---|---|---|---|---|---|---|
| 140 | 3.7 | 64 | 2.7 | 64 | 2.7 | 2.7 |
| 150 | 3.8 | 76 | 3.5 | 78 | 7.5 | 11.5 |
| 160 | 3.2 | 64 | 3.4 | 73 | 17.0 | 25.2 |
| 170 | 2.7 | 56 | 3.7 | 67 | 19.4 | 31.0 |

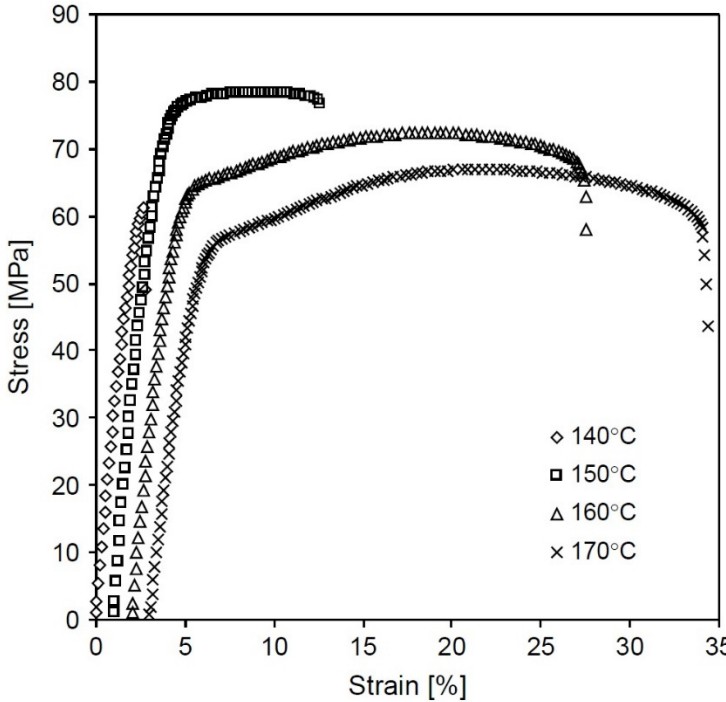

**Figure 7.** Typical stress–strain curves at various polymerization temperatures for dry as-molded (DAM) APA-6 (curves shifted to the right for clarity). Reprinted from Reference [10], under open-access license.

If, on the other hand, the temperature decreases, the degree of crystallization in-creases, but this does not lead to an increase in strength and rigidity; this is because, as mentioned previously, it is caused by the fact that the crystallization rate is so high that it retains the caprolactam monomers within the crystallites, thus increasing the fragility of the material.

In addition to the monomer, even the reactive end parts of the chains can remain blocked; consequently, not only is there an abrupt interruption of the growth of the macro-molecule, but the interdiffusion of the chains between the amorphous and crystalline phases is also limited [10].

5.2.7. Effect of the Polymerization Temperature on the Polymer Density and Void Content

The density of the crystalline phase ($\rho = 1.24$ g/cm$^3$) is greater than the density of the amorphous phase ($\rho = 1.08$ g/cm$^3$) [83], and, consequently, a lower degree of crystallinity leads to obtaining a lighter polymer.

As expected, going down from 170 to 150 °C, there is an increase in density because the degree of crystallization increases, see Figure 8. The crystallization process results in about 9% shrinkage [92], and the higher the degree of crystallization, the higher the degree of shrinkage. The mold heats the reactive mixture from the outside to the inside, and the polymerization proceeds in the same direction; crystallization will occur in the final stages of polymerization, and for this reason, the shrinkage is initially compensated by a low-viscosity caprolactam flow, resulting in a polymer with no internal voids.

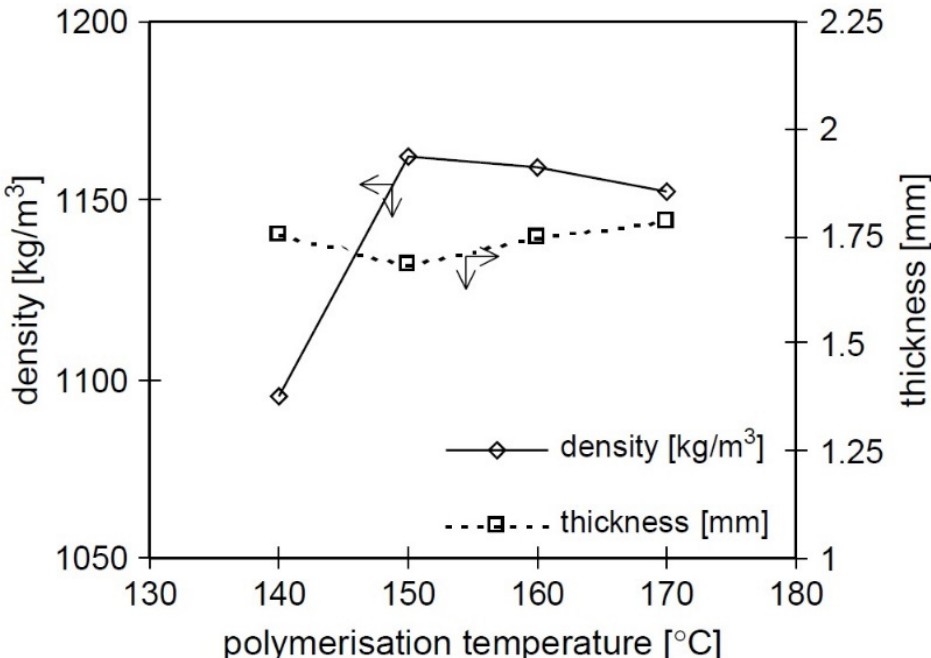

**Figure 8.** Density and panel thickness for various polymerization temperatures. Reprinted from Reference [10], under open-access license.

At 140 °C, however, the density suddenly drops just when the crystallinity is greater. This behavior can help in explaining the lower mechanical properties found: it is assumed that, at this temperature, the crystallization speed and, therefore, the volume contraction do not allow the liquid flow of caprolactam to keep up with the crystalline phase being formed. In this way, the volumetric shrinkage does not affect the external dimensions of the product but leads to the formation of internal voids, see Figure 9. The presence of voids undermines the mechanical characteristics and leads to a decrease in density. When voids are formed, the internal pressure is lowered until it falls below the evaporation pressure of the caprolactam; the voids will therefore be filled with caprolactam vapors, which, however, do not take part in the polymerization reaction, also causing a decrease in the degree of conversion [10].

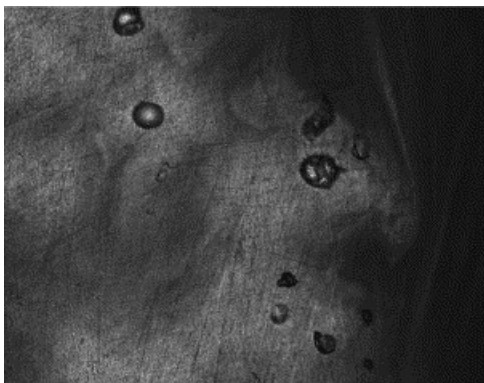

**Figure 9.** Optical micrograph of the cross-section of APA-6 polymerized at 140 °C. Reprinted from Reference [10], under open-access license.

It is necessary to work at an optimal temperature that considers the various effects described above. The best mechanical performance is obtained for that temperature value for which there is an absence of internal voids, but, in any case, with a high number of intact crystals without internal inclusions of caprolactam. However, it must be remembered that, even if the optimal polymerization temperature is found and set as such, the polymerization and crystallization reactions are exothermic, and therefore the temperature inside the mold will inevitably tend to increase. For this reason, the influences of other parameters, such as the geometry of the mold, the formulation used for the reactive mixture, and the quantity of fibers used as reinforcement, will also become important.

As for composite materials, the presence of fiber reinforcement helps the crystallization process thanks to both its nucleating effect and the partial absorption of the heat generated by the reaction. However, the fibers hinder the movement of the low-viscosity caprolactam flow, potentially causing shrinkage-induced voids to form even at higher temperatures [10].

### 5.2.8. Effect of Demolding Time on Mechanical Properties

Thanks to the study of van Rijswijk et al. [8], the effect of the demolding time on the properties of the final polymer was studied. To obtain APA-6 specimens, hexameth-ylene-1,6-dicarbamoylcaprolactam (C20) was chosen as the activator and caprolactam magnesium bromide (C1) as the initiator. In this study, two demolding times of 30 and 45 min were compared for various polymerization temperatures. From Figures 10 and 11, it is possible to see how polymerization times of 45 min have allowed for the achievement of better mechanical properties. It must be remembered that, for temperature values between 140 and 150 °C, the best characteristics in terms of crystallinity are obtained, as already described above, and that is why the highest elastic-modulus and tensile-strength values are present.

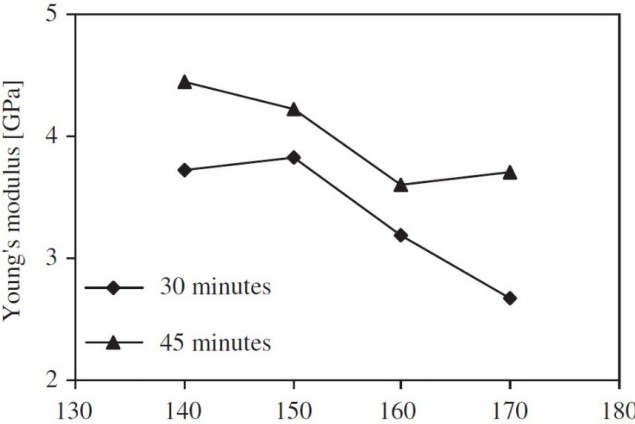

**Figure 10.** Young's modulus at 23 °C for various mold temperatures and for various polymerization times. Dry as-molded properties. Reprinted from Reference [8], under open-access license.

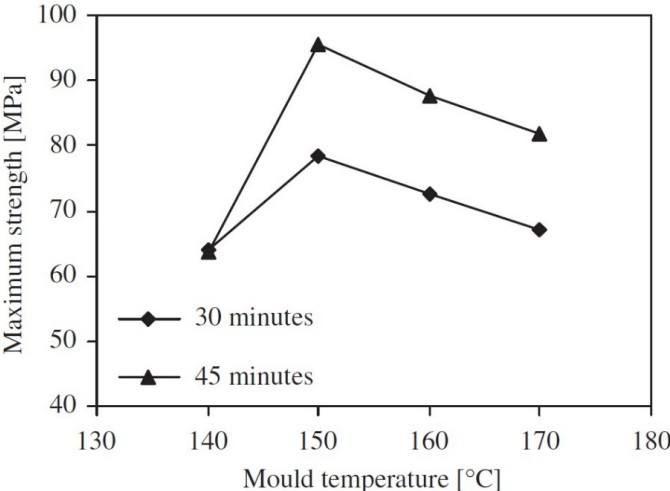

**Figure 11.** Maximum strength 23 °C for various mold temperatures and for various polymerization times. Dry as-molded properties. Reprinted from Reference [8], under open-access license.

5.2.9. Thermal Interaction between Fiber and Matrix

The addition of a volumetric percentage of fibers has a significant effect on heat transfer during the infusion process. The presence of reinforcement facilitates the in-crease of the reaction speed by reducing the time available for the infusion. This is because the fibers placed inside the mold are preheated together with it, and, therefore, the resin that propagates inside them during the infusion will heat up faster, since the total conductivity is increased thanks to the presence of the fibers [44].

Figure 12 shows the drop in the temperature gradient inside the mold that occurs immediately after the start of the infusion.

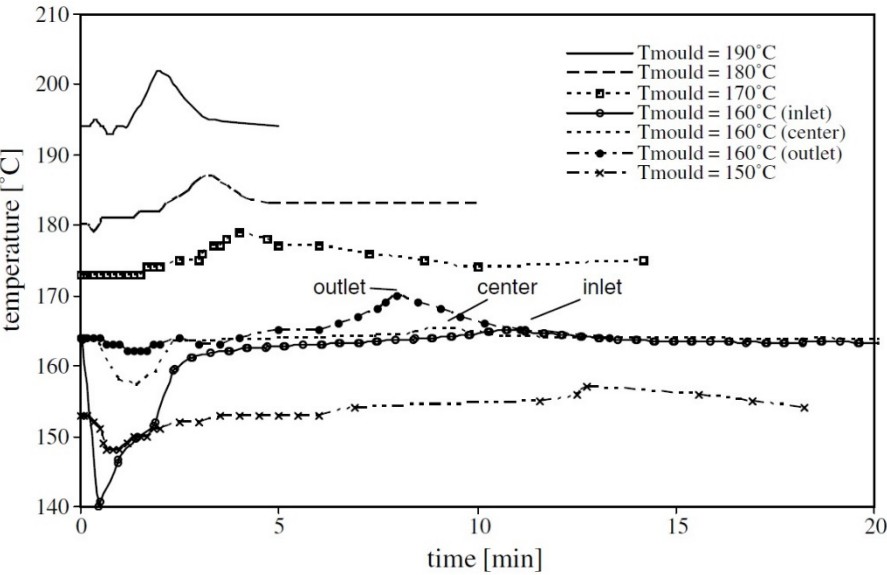

**Figure 12.** Temperature progression during processing APA-6 composites for various initial mold temperatures. At the start of the infusion, t = 0 s. Reprinted from Reference [44], under open-access license.

The reason is the following: compared to the temperature of the mold and the fibers inside it (140–180 °C), the resin is at a lower temperature (110–130 °C), thus causing an instant cooling of the fibers present. Continuing the infusion, the curve rises again as the resin increases its temperature, bringing itself into equilibrium with that of the mold, and then remains approximately constant. What occurs is that the resin temperature, during

the impregnation phase, will assume the lowest values at the point near the entrance, and highest at the point near the exit from the mold. This means that the resin present in the outlet will have a longer residence time than the resin present in the inlet, and, therefore, the polymerization will start right from the final part of the mold. The reaction front will therefore be inverse with respect to the infusion front. For this reason, the peak of the exothermic reaction occurs, for example, in the case of the mold temperature equal to 160 °C after 8, 9, and 11 min, starting from the outlet, center, and inlet [44].

Generally, in the presence of the matrix alone, there is an increase in the temperature of the exothermic reaction of the order of 10–30 °C [8], based on the reaction rate and thickness of the mold; for a composite, on the other hand, this increase is even less than 5 °C. This is because the presence of the fibers implies an occupation of the mold that will not be destined for the resin, causing a reduction in the quantity of reactive mixture that can enter it and, therefore, a reduction in the exothermic reaction, as can be seen from the reduction of the polymerization ΔH and crystallization in Table 3.

**Table 3.** Thermal properties. Reproduced from Reference [44].

| Material | Density (kg/m$^3$) | Conductivity (J/(s m °C)) | Specific Heat (J/(kg °C)) | ΔHpol (J/g) | ΔHcryst (J/g) |
|---|---|---|---|---|---|
| Caprolactam | 950 | 0.14 | 2500 | – | – |
| Anionic polyamide-6 | 1130 | 0.2 | 2250 | 166 | 144 |
| Glass fiber | 2525 | 1 | 840 | – | – |
| Composite | 1760–1912 | 0.57–0.6 | 1670–1545 | 47 | 41 |

### 5.2.10. Interaction at the Interface between Fiber and Matrix

To obtain the optimal mechanical properties of composite material, it is necessary to ensure a strong bond between the fiber and the matrix at the interface. In general, fiber fabrics are intended to produce composite materials with thermosetting matrices, making their application and interaction with thermoplastic matrices more complicated. Carbon fiber fabrics are almost always intended to produce composites with epoxy matrices resulting in little favor of the formation of an interfacial bond with a thermoplastic matrix, such as polyamide. It is a broad and complex topic that requires an in-depth study of which type of sizing is suitable for a specific thermoplastic matrix.

Van Rijswijk et al. [93] have studied the application for an improvement of the type of coupling agent that promotes a bonding reaction between fiber and matrix by aminosilane groups that are compatible with the Nylon-6 polyamide matrix obtained by polymerization in situ. These groups will react with the glass surface through a condensation reaction, and, subsequently, through a further condensation reaction between amino silane molecules, branching occurs [44], see Figure 13.

**Figure 13.** Sizing of glass fibers with aminosilane coupling agents. Reprinted from Reference [44], under open-access license.

In their study, it emerged that aminosilanes form an adequate bond with the thermoplastic matrix, without incurring the problem of termination of the polymer chains: after the unblocking of the activator and release of the isocyanate groups, the latter react with the amino group of the amino silane, obtaining a urea bond, see Figure 14.

**Figure 14.** De-blocking of carbamoylcaprolactam (**Step 1**) and the subsequent formation of branch points (**Step 2a**) or fiber-to-matrix bonds (**Step 2b**). Reprinted from Reference [93], under open-access license.

### 5.2.11. Effect of the Polymerization Temperature on the Properties of the Composite

The following figures show the curves of the degree of crystallization and conversion as a function of the polymerization temperature. For the graphs in question, van Rijswijks studied the behavior of APA-6 matrix composites obtained with 1.2 mol% difunctional hexamethylene-1,6-dicarbamoylcaprolactam (HDCL) and 1.2 mol% caprolactam magnesium bromide (MgBrCL), and with two types of E-glass fabric (300 g/m$^2$): the first without the presence of sizing, and the second with aminosilane sizing. Since the exothermic effect is lower due to the presence of fibers, it is necessary to increase the temperature of the mold to prevent the reactive end parts of the chains from being trapped inside the rapidly forming crystals when working at low temperatures. From both graphs, it is possible to note that a good choice for a polymerization temperature is that equal to 160 °C: for this value, there is a degree of crystallization for the composite that is greater than that of the matrix alone, and a degree of conversion that is similar.

In the event of an increase in the temperature of the mold, the degree of crystallization decreases, and toward 170 °C, the curves undergo an inversion, showing how the composite will go toward lower degrees of crystallization than the matrix alone, since the interactions of the matrix with the fibers, making crystal formation more complicated [44]. To show a practical example, Yan et al. [86] investigated the effects of activator variation, polymerization time, and polymerization temperature on mean molar mass, degree of crystallization, and mechanical properties. In this specific case, the PA-6 composite was studied and obtained by anionic polymerization in situ by vacuum infusion, with glass-fiber reinforcement. For the GF/APA-6 composite, 400 g/m$^2$ S-glass was used as fabric reinforcement, sodium caprolactamate (C10) as initiator, and hexamethylene-1,6-dicarbamoylcaprolactam (HDCL, C20) as activator. Both the activator and initiator were dried at 50 °C for 24 h, while the fabric was dried for 24 h at 120 °C. Figure 15 shows the trend of the average molecular mass and the degree of crystallization for various polymerization temperatures, keeping the activator content constant equal to 1 mol% (C10:C20 equal to 2:1) and a polymerization time of 60 min.

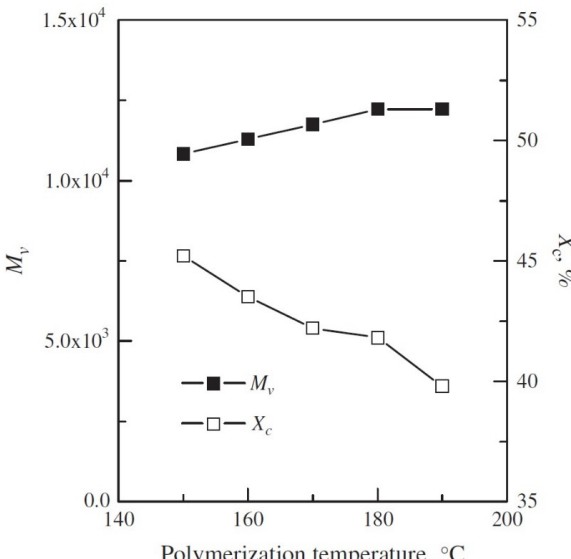

**Figure 15.** Effects of polymerization temperature on the viscosity average molecular weight and degree of crystallinity of the matrix resin of GF/APA-6 composites. Reprinted from Reference [86], under open-access license.

As expected, as the polymerization temperature increases, the polymerization rate increases, obtaining a higher molecular-weight value. However, it will be lower than the value of the APA-6 resin alone for the reasons already described above: the absorption of the reaction heat by the fibers, leading to the reduction of the reaction temperature; and the increase of the crystallization reaction, which limits the polymerization reaction and presence of the groups that are reactive on the surface of the fibers, such as acid groups and hydroxyl groups, that end the growth reactions of macromolecules.

The degree of crystallization also decreases precisely because the branching process is favored at higher temperatures, and this hinders the crystallization process.

As for the mechanical properties, the tensile strength increases from 363 to 434 MPa as the polymerization temperature increases, reaching a maximum at 180 °C, while the elastic modulus varies slightly [86], see Figure 16.

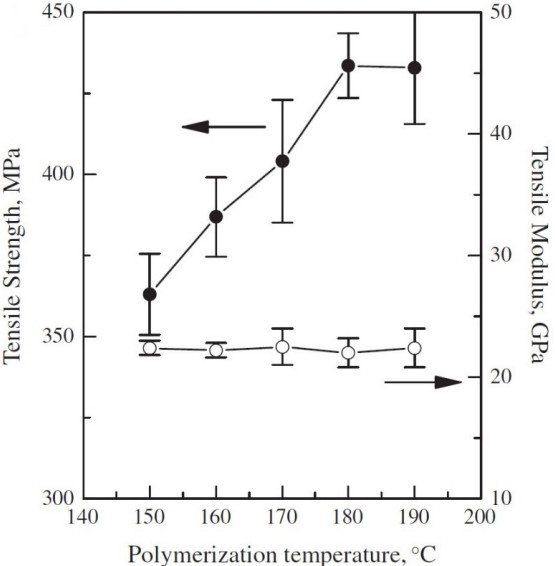

**Figure 16.** Effects of polymerization temperature on the mechanical properties of GF/APA-6 composites. Adapted from Reference [86], under open-access license.

*5.3. Examples of Fiber-Reinforced Thermoplastic Composites Currently Available Commercially*

Researchers and industry are currently working toward a circular economy, which is an industrial system aimed at reusing, remanufacturing, and recycling of products at their end-of-life (EOL) stage. When reusing, remanufacturing, and further reselling of products is impracticable, closed-loop recycling is exercised to retain materials in use and circulation [94]. Overall, the circular economy aims at using renewable energy, eliminating toxic chemicals, and eliminating waste by better design of materials, products, systems, and business.

In light of these considerations and with a view to obtaining more sustainable composites, it is currently the Johns Manville company that is working on this front, producing performing composites with excellent mechanical properties. It is a leading manufacturer and distributor of high-quality insulation and commercial roofing, along with fiberglass and nonwovens for commercial, industrial, and residential applications [95]. Specifically, OS-6 and NCF-6 series organosheets, shown in Figure 17 (left), are produced through impregnation and in situ polymerizations of caprolactam. On the right, there is an SEM image showing the full impregnation of the fibers by Nylon 6 that permits us to obtain a high thermal and mechanical performance.

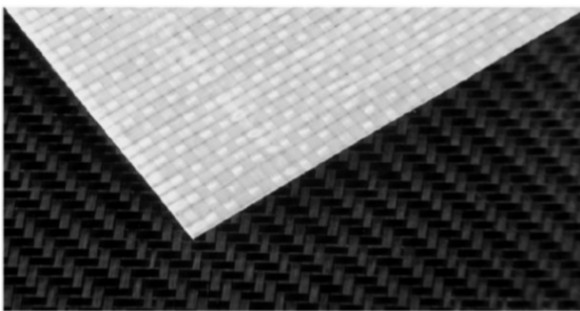 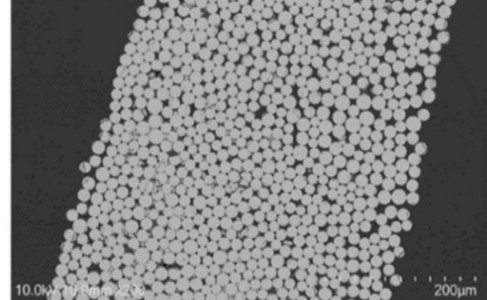

**Figure 17. Left**: Glass fiber and carbon fiber OS-6 series Nylon-6 organosheets. **Right**: Fully impregnated Nylon-6 organosheet [90].

## 6. Environmental Impact

### 6.1. Life-Cycle Assessment

The evolution of the production processes of polymer-based materials has made them applicable to a very wide variety of sectors. The total world production of polymers in recent decades has attracted attention to the issue of their disposal at the end of the useful life cycle of the products of this industry. In line with the goals of the European Union, in this review, in addition to the technical aspect of the production processes, the environmental aspect was also analyzed. Currently, the most recommended tool for the quantitative analysis of a process or product is the life-cycle assessment (LCA), promoted and constantly updated by the United Nation Environment Program (UNEP). UNEP annually appoints specific task forces for each environmental and cultural sector to define the methods for assessing environmental impacts, specific indicators by impact category and protection area [96], and for data collection [97]. This last phase takes place in synchrony with the most developed industrial sectors which provide detailed datasets on specific processes. The variety of data to be collected requires appropriate organizations divided by industrial sector or very often by type of material or production process. In the case of composites, this task is entrusted to the European Composites Industry Association (EuCIA), which has an environmental impact calculator based on UNEP guidelines [96]. This tool, which is accessible through the EuCIA website in the ECO IMPACT CALCULATOR section, has datasets for specific production processes of composites with different polymeric matrices. Through the methods suggested by UNEP, this eco-tool reports the environmental impact related to the production of a certain mass of composite, depending on the masses of the materials used and the specific production process [98]. For the purposes of this review,

the impacts of the most common production processes, related to the same composite produced, were calculated and compared by using the ECO IMPACT CALCULATOR.

### 6.1.1. Goal and Scope

The goal of this analysis is to compare the environmental impacts of different production processes related to the same type of composite through an LCA screening. A functional unit of one kilogram of the composite produced was set to obtain specific results on the unit of mass. The analysis proposed by the software is of the cradle-to-gate type; that is, the system boundaries include only the phase of finding the raw materials and their transformation into a finished product. Excluding the life of the product and final disposal, the resulting impacts are lower for more easily available materials and processes with lower consumption of materials/energy and with lower emissions. For a broader comparative analysis, the impacts related to the production of a thermoplastic matrix composite (PA-6) with three different production processes were first assessed: pultrusion, injection molding (IM), and resin transfer molding (RTM). Finally, for the latter process, the impacts related to the production of a kilogram of two identical composites were assessed by varying only the polymer matrix, PA-6, and epoxy resin. The second comparison, all other parameters being equal, refers exclusively to the impacts due to the production of the starting monomer and to the polymerization/crystallization process of the two major exponents of thermoplastic and thermosetting polymers.

### 6.1.2. Inventory

The ECO IMPACT CALCULATOR evaluates materials and processes with reference to the European databases of Ecoinvent [99], European reference Life Cycle Database (ELCD) [100], and those developed by Plastics Europe [101]. The data were aggregated in inputs and outputs in the categories of use of resources, emissions into the air, and discharges into water and soil. For all the composites assessed, a composition by mass of 60% of glass fiber assembled roving, and 40% of the polymer was considered, without considering other additives. The datasets referring to the three processes considered are the default ones of the software and include the flows of matter and energy into and out of the previously defined system boundaries.

### 6.1.3. Life Cycle Impact Assessment (LCIA)

For each composite-process configuration that was analyzed, the environmental impacts were assessed by using two globally recognized methods:

- Greenhouse Gas Protocol V1.01/$CO_2$ eq (kg);
- Cumulative Energy Demand V1.09/CED (MJ).

The Greenhouse Gas (GHG) Protocol was defined by a multi-stakeholder partnership of businesses, non-governmental organizations (NGOs), governments, World Resources Institute (WRI), and the World Business Council for Sustainable Development (WBCSD) [98]. This method defines the carbon footprint of the production process considered. The indicator used to express the contributions of different chemical species emitted to earth, water, and soil is unique and normalized for all effects, and it is calculated in kilograms of equivalent carbon dioxide. The CED is the total amount of energy resources needed to produce the composite and includes energy from renewable and non-renewable sources. The indicator associated with this method is the CED, and it is expressed in Megajoules [98].

Figure 18 presents the results of the calculation carried out with ECO IMPACT CALCULATOR in which it is clear that, for both the GHGs and the CED, the most advantageous process is the RTM. When analyzing the inventory of the different processes, we see that the lower impact of the RTM compared to the other two processes is due to the non-use of water and the much smaller amount of electricity required. Compared to the other two processes, in fact, the RTM equipment does not include rotating elements and large movements of viscous fluids. Furthermore, the lower hazardous emissions contribute to making it the greenest process among the three.

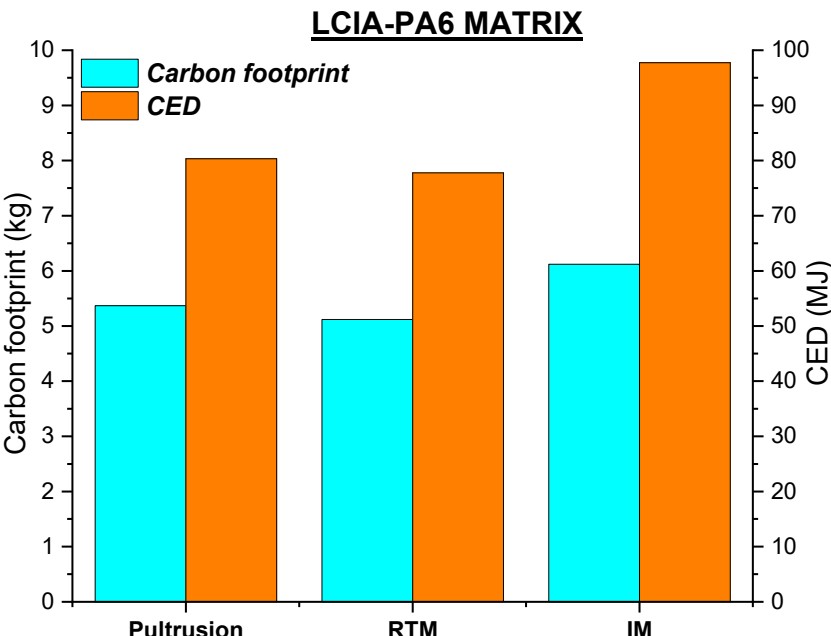

**Figure 18.** LCIA results related to three composite manufacturing processes: pultrusion, injection molding (IM), and resin transfer molding (RTM). The carbon footprint is indicated on the left ordinate, and the cumulative energy demand (CED) is on the right ordinate.

Considering the RTM process, the impact of a type of composite with a different polymer matrix was evaluated. In this case, as shown in Figure 19, the least impact is obtained by using an epoxy resin instead of PA-6. This result is justified by the use and development, over the years, of equipment optimized for thermosetting resins which are not fully adaptable to the characteristics of thermoplastics; this, in any case, makes them only slightly more impactful than resins in regard to the production of finished composites.

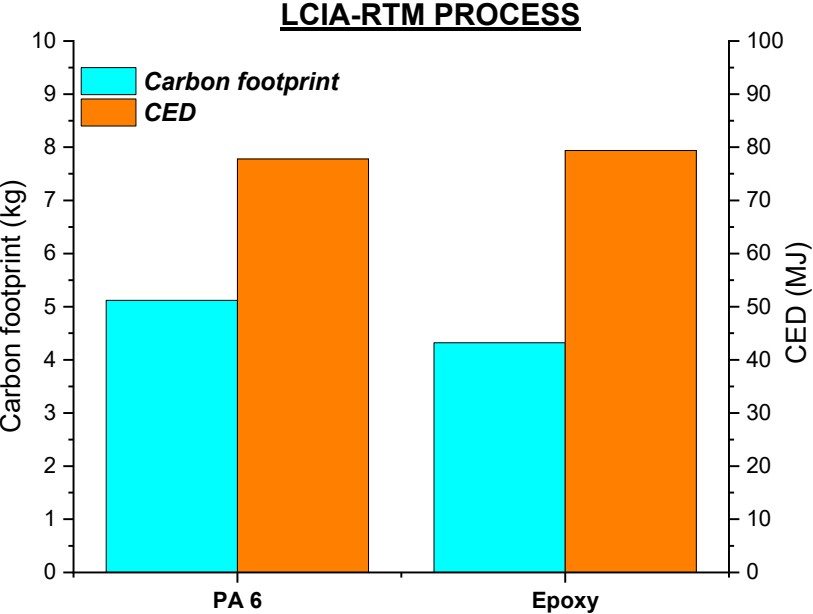

**Figure 19.** Results of the LCIA relative to two composites with different polymeric matrices: PA6 and epoxy resin. The carbon footprint is indicated on the left ordinate (blue), and the cumulative energy demand (CED) is on the right ordinate (orange).

### 6.2. Limitation

The environmental impact assessment carried out refers only to the procurement phases of raw materials and the transformation processes into the finished product. The phase of the product's life was excluded from the analysis and, above all, from the final disposals. Going deeper into this last phase, it is possible to get an idea of the cyclic nature of the material used, and, especially in the comparison between the epoxy resin and PA-6, it could turn in favor of thermoplastics [102]. The disposal phase involves three different scenarios: reuse, recycling, and disposal. Due to their chemical nature, thermoplastics follow the path of recycling in most cases. However, among the most recent studies, a solution has emerged for the reuse of reinforcements of thermosetting matrix composites for subsequent use with thermoplastics [103]. Furthermore, the possibility of using recyclamines for the recycling of thermosetting matrices has been analyzed and is still under development [104]. These scenarios, evaluated through LCA, could lead to more detailed analysis and, therefore, to targeted technical choices that include the entire life cycle of the materials used.

For a broader view of the comparisons between composite production techniques, an analysis of processes with "in situ" polymerization would be necessary. This technique involves the use of precursors that are at an earlier level in the supply chain than the use of polymers, and this would make them competitive in terms of environmental impact. Finally, the lack of a large number of specific datasets for vacuum infusion still makes it impossible to aggregate consistent and reliable data for the construction of an inventory referring to this process.

### 7. Conclusions

In the present article, the most reported techniques to produce composite materials and how some process variables play a fundamental role were reviewed, focusing, above all, on the feasibility of the latter for thermoplastic matrix composites. There are two fundamental lines to focus on: (I) the production process and the chemistry of the matrices to obtain a performing thermoplastic matrix composite, with mechanical–functional characteristics suitable for completely replacing thermosetting agents; and (II) the use, recovery, and disposal of thermosetting and thermoplastic composites which re-enter the circulation to obtain manufactured articles. These increase the concept of circularity. The big problem is that thermoplastic matrix composites are almost never performing compared to thermosetting agents, due to their poor impregnability in the production of composites with a high fiber content; indeed, still, almost all of these materials are thermosetting. The reason lies in the chemical and, above all, physical nature of thermoplastics. They never have a fluidity phase, which is why very high temperatures are required, with the consequence of degrading the polymer. It is therefore not possible to use too-high process temperatures, and, in any case, in the event of a compromise temperature, there would still be the problem of a high viscosity. This leads to extremely defective composites with a high fiber content and, overall, low-quality composites.

The solution lies in in situ polymerization precisely because it no longer allows starting from a polymer, in which macromolecules are already formed, which results in high viscosity and process temperatures around 200 °C, but from monomers that have a similar viscosity to that of water and which therefore allow a process not too far from those already known for thermosetting. Vacuum infusion turns out to be one of the most effective techniques that allows to treat high volumes with simpler and cheaper equipment. However, it will be necessary to move toward an optimization of the process parameters and, consequently, of the impregnation phase, guaranteeing a greater homogeneity of dispersion of the reactive mixture and a better adhesion between fiber and matrix, aiming at the study of sizing compatible with thermoplastics matrices. These resolutions will lead to the real goal, which is the production of performing composite materials with a thermoplastic matrix, with increasing volumetric quantities of reinforcement, up to values equal to 40%, allowing a true comparison, in terms of both the quantity of filling

and mechanical properties, with composite materials with a thermosetting matrix. This review work was carried out as our research group is working on the implementation and optimization of the infusion technological process of tissue and, especially, TNT reinforcing structure with the aim of obtaining thermoplastic composites reinforced with high amount of fiber reinforcement as less impregnation defect. Its impregnation is very difficult with current conventional techniques but, the vacuum infusion, or assisted from secondary pressure step, if operate with low viscosities matrix materials inevitably leads to a better impregnation.

**Author Contributions:** Conceptualization, M.V.; data curation, M.V. and M.S.; writing—original draft preparation, I.R. and I.B.; writing—review and editing, M.V. and M.S.; supervision, M.V. All authors have read and agreed to the published version of the manuscript.

**Funding:** This research received no external founding.

**Acknowledgments:** The research group under the guidance of Marco Valente would like to thank the companies, companies Bruggeman GmbH & Co. KG (Ap-Nylon) and Johns Manville Advanced Composites. for their contribution to the work, both in terms of information and material given for free.

**Conflicts of Interest:** The authors declare no conflict of interest.

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
