# Peer review of "Thermoplastic Composite Materials Approach for More Circular Components: From Monomer to In Situ Polymerization, a Review"

_jcs, doi:10.3390/jcs6050132_

Round 1
Reviewer 1 Report
This article reviews on the thermoplastic composite materials approach for more circular components from monomer to in situ polymerization. Special attention is focused on the anionic polymerization, in situ polymerization for PA6 and composites, vacuum infusion molding and environmental impact. It is well-structured and with novelty. I recommend it accepted after minor revision. The specific points are as follows:
1. The length of sub-section in the following sections should be comparable, such in 3. In situ poly merization, 4. In situ polymerization for composites, and 5 Vacuum infusion molding.
2. The resolution of figures should be improved and some figures are shown without the top part, such as Figure 3, 4, 7, etc.
3. It is generally not need to be abbreviated for titles, for example 6.1 LCA.
Author Response
Cover letters for reviewer
Reviewer 1
The authors would like to thank the reviewer for his valuable comments and suggestions on this manuscript. Below are our replies to comments
- The length of sub-section in the following sections should be comparable, such in 3. In situ polymerization, 4. In situ polymerization for composites, and 5 Vacuum infusion molding.
1.r According to the reviewer’s comment, section 3 (lines 164-172, 184-190, Table 1) and 4 (lines 267-295) have been implemented in order to make them comparable. Section 5 cannot be reduced as it contains the central theme of the review.
- The resolution of figures should be improved and some figures are shown without the top part, such as Figure 3, 4, 7, etc.
2.r According to the reviewer’s request, the image quality has been improved.
- It is generally not need to be abbreviated for titles, for example 6.1 LCA.
3.r At the request of the reviewer, the title of subsection 6.1. was changed from "LCA" to "Life Cycle Analysis".
Reviewer 2 Report
I my opinion, the review paper entitled "Thermoplastic composite materials approach for more circular components: from monomer to in situ polymerization, a Review" should be published as the journal article.
The topic of the presented manuscript is very actuall and discussed important topic of the development of new materials for composite processing. Some minor correction should be made do improve the English quality, example: Fig.6 where "cristallinity" should be corrected. However, this type of correction may be made at the stage of proof preparation.
In my opinion, the text could be supplemented with some basic information about the standard PA6-based composites produced by injection molding. Such a paragraph could better explain the advantages of using fabric reinforcement over short fibers .
Author Response
Cover letters for reviewer
Reviewer 2
The authors would like to thank the reviewer for his valuable comments and suggestions on this manuscript. Below are our replies to comments
- The topic of the presented manuscript is very actuall and discussed important topic of the development of new materials for composite processing. Some minor correction should be made do improve the English quality, example: Fig.6 where "cristallinity" should be corrected. However, this type of correction may be made at the stage of proof preparation.
1.r At the request of the reviewer, the use of English was rechecked, and some corrections were made, including the caption of Figure 6.
- In my opinion, the text could be supplemented with some basic information about the standard PA6-based composites produced by injection molding. Such a paragraph could better explain the advantages of using fabric reinforcement over short fibers
2.r According to the reviewer’s request, the paragraph has been implemented: starting from line 267 up to 295.